# Theoretical Study of the Exciton Binding Energy and Exciton Absorption in Different Hyperbolic-Type Quantum Wells under Applied Electric, Magnetic, and Intense Laser Fields

**DOI:** 10.3390/ijms231911429

**Published:** 2022-09-28

**Authors:** Melike Behiye Yücel, Huseyin Sari, Carlos M. Duque, Carlos A. Duque, Esin Kasapoglu

**Affiliations:** 1Department of Physics, Faculty of Science, Akdeniz University, Antalya 07058, Turkey; 2Department of Mathematical and Natural Science, Faculty of Education, Sivas Cumhuriyet University, Sivas 58140, Turkey; 3Grupo de Materia Condensada-UdeA, Facultad de Ciencias Exactas y Naturales, Instituto de Física, Universidad de Antioquia UdeA, Calle 70 No. 52-21, Medellín AA 1226, Colombia; 4Department of Physics, Faculty of Science, Sivas Cumhuriyet University, Sivas 58140, Turkey

**Keywords:** exciton binding energy, exciton absorption, electric and magnetic field, intense laser field

## Abstract

In this study, we investigated the exciton binding energy and interband transition between the electron and heavy-hole for the single and double quantum wells which have different hyperbolic-type potential functions subject to electric, magnetic, and non-resonant intense laser fields. The results obtained show that the geometric shapes of the structure and the applied external fields are very effective on the electronic and optical properties. In the absence of the external fields, the exciton binding energy is a decreasing function of increasing well sizes except for the strong confinement regime. Therefore, for all applied external fields, the increase in the well widths produces a red-shift at the absorption peak positions. The magnetic field causes an increase in the exciton binding energy and provides a blue-shift of the absorption peak positions corresponding to interband transitions. The effect of the electric field is quite pronounced in the weak confinement regime, it causes localization in opposite directions of the quantum wells of the electron and hole, thereby weakening the Coulomb interaction between them, causing a decrease in exciton binding energy, and a red-shift of the peak positions corresponding to the interband transitions. Generally, an intense laser field causes a decrease in the exciton binding energy and produces a red-shift of the peak positions corresponding to interband transitions.

## 1. Introduction

The primary purpose in selecting the most appropriate potential when investigating low-dimensional quantum systems is to best reflect the atomic structure of the material of interest and to provide the design of new optoelectronic devices by manipulating the electronic and optical properties. In this context, the shape of the confinement potential is very important for the manufacture of advanced devices. Analytical modeling is widely used in many areas of physics. Analytic models are employed to establish paradigms, or to form the main properties of a physical system. For example, the harmonic oscillator problem is widely used in many areas of physics, such as quantum optics, solid state physics, or molecular physics, to describe modes of the electromagnetic field, crystal lattice, or molecular vibrations. The deficiency of analytically solvable models in physics led to the development of quasi-exactly solvable analytical models.

Apart from fully solvable potentials by using many different approximation methods such as the hydrogen atom, quantum harmonic oscillator, and pseudoharmonic and Mie-type potentials [1,2,3,4], many so-called quasi-exactly solvable (QES) potentials, such as Manning–Rosen [5,6], Hultheen [7], Morse [8], Razavy [9,10], generalized Pöschl–Teller [11,12], and Kratzer-type [13] potentials, are very useful in describing the potential energies of the diatomic or polyatomic molecules. The use of the hyperbolic-type potentials, which in this work depends on the adjustable parameters, is an excellent strategy to implement the non-abrupt variations of the potential in the interfaces of the materials that make up the heterostructures, which in real systems have a width of a few tenths of a nanometer. Using such types of potential models is possible to account for the electro- and magneto-optical properties of isolated and coupled double quantum wells (DQW) with finite potential barriers and explain the interdiffusion phenomena in multiple quantum wells (QW) [14].

Exact solutions of QES-type potentials have been generally obtained by using the phenomenological potentials in the form of polynomial [15], exponential [16], trigonometric [17], and hyperbolic-type functions [18]. As known, the phenomenological potentials are obtained by using approaches such as selecting a mathematical function and fitting its unknown parameters to some properties, experimentally determined, of the material. Thanks to advances in technology, growth techniques have a high degree of precision and control over the characteristics, such as the size and shape of the semiconductor heterostructures to be formed. Before the growth of any low-dimensional quantum structure, investigating by choosing suitable phenomenological potential for the structure to be studied is very efficient in terms of both cost and time. The more the realistic structure of the confinement potential on the electronic and optical properties of low-dimensional systems (LDS) is important, the more the effects of the applied external fields are important. Thus, these potentials mentioned above have been extensively investigated under external fields such as electric, magnetic, and intense laser fields (ILFs). The effects of the ILF on the electronic structure in a Gaussian quantum well were investigated by Sari et al. [19]; Kasapoglu et al. investigated the effects of non-resonant high-frequency ILF on the electronic and optical properties of both the symmetric and asymmetric double Morse quantum wells, and quantum wells/quantum dots which have Razavy potential [20,21]. Furthermore, some of these potentials, which are called QES, have been studied under electric and magnetic fields [22,23,24,25,26,27].

It is an undeniable fact that the transition from bulk structure to LDSs such as quantum well, quantum wire, and quantum dots causes an increase in Coulomb interaction and more strongly bound electron–hole pairs (excitons). Since the excitons are confined in a narrower plane than their Bohr radius in most semiconductor heterostructures, quantum confinement enhances the exciton binding energy (EBE), altering the wavelength of light they absorb and emit. These physical facts cause excitons to bind even at room temperature [28]. In this context, LDSs make the excitons easily tunable, with a variety of external fields and different sample geometries, enabling them as potential candidates for various applications in optics and optoelectronics. The changes in physical properties caused by the reduction in size make these systems promising candidates for next-generation electronics and optoelectronics [29]. By using the Casida equation formalism for molecular excitations to periodic solids, Yang et al. obtained the exciton binding energies directly [30]. Particularly, they calculated the exciton binding energies for both small- and large-gap semiconductors and insulators. Weerasinghe and coworkers used a coherently degenerate state strategy to design and synthesize an organic small molecule dimer compound consisting of two triphenylamine-based monomers [31]. The time-resolved fluorescence measurements showed that the lifetime of the Wannier excitons in the newly designed dimer compound is less solvent-dependent than that of the Frenkel excitons in the monomer. In addition, the authors showed that the power conversion efficiency of the photovoltaics constructed using this new material increases by eight-fold in comparison to that using the monomer.

In recent years, semiconductor growth technology has advanced to such an extent that it is now possible to grow thin-layered semiconductor structures of very high quality. These layers can be controlled so thin that the particle confinement of electrons and holes in a box is easily possible [32,33]. For instance, with the GaAs/AIGaAs material system, because the larger band-gap AlGaAs “barriers” have both lower valence-band edges and higher conduction-band edges than the GaAs, the alternate thin layers result in confinement of both electrons and holes within the GaAs layers. Consequently, excitons may also be confined within the GaAs layers. If the AlGaAs barriers are sufficiently thick and have a sufficiently large Al concentration so that the potential barriers are high, then penetration of the wavefunctions from one GaAs layer (or quantum well) to another may be neglected at least for the low-energy states within the quantum well; then we may refer to the structure as a multiple-quantum-well structure (MQWS). The physics of such an MQWS for these low-energy states is, for many purposes, the physics of a single quantum well, although it is convenient for optical-absorption studies to grow an MQWS to obtain sufficient optical absorption [33]. Another important feature of these systems that should be emphasized is that in multiple-quantum-well structures, the exciton resonances can be clearly observed at room temperature [34,35,36]. This phenomenon is remarkable because in most bulk semiconductors, the exciton resonance is all but unresolvable at room temperature. The observation is of potential practical importance, as this property of low-dimensional semiconductor heterostructures allows the development of room-temperature optical devices using excitons. On the other hand, in comparison with conventional electronic devices, excitonic devices show great potential in efficient signal processing at a higher speed. Excitons can be used for signal processing and simultaneously can directly link to optical communication. Therefore, the delay between optical communication and signal processing can be effectively eliminated in excitonic devices, which possess a significant advantage in high-speed communications [37,38,39].

The present work is concerned with the theoretical study of the EBE and interband excitonic transitions in single and double QWs represented by hyperbolic functions of the spatial coordinate under the electric field, magnetic field, and non-resonant ILF. The paper is organized as follows: Section 2 contains the theoretical description; the obtained results are discussed in Section 3; finally, the conclusions are given in Section 4.

## 2. Results and Discussion

Other than those given in the theory section, the values of material parameters of GaAs/AlGaAs used in our calculations are (with x=0.3 for the aluminum concentration in the barrier regions) $m_{e}^{*}=0.067\;m_{0}$, $m_{h}^{*}=0.45\;m_{0}$ (where $m_{0}$ is the free electron mass), ε=12.5, Γ=1.0 meV, $E_{p}=25.7$ eV, and $E_{g}^{GaAs}=1.424$ eV [40,41].

Heavy-hole EBEs as a function of the well width (*k* parameter) for confinement potential $(V_{1}\left(z_{e(h)}<0\right)$ in Equation (Equation 2) (see the next section) under the external applied fields such as electric field, magnetic field, and ILF are given in Figure 1a–c, respectively. Results are given for $\nu_{1}=1$. In the absence (solid lines) and presence of the external fields (dashed and dot lines), confinement potential profiles for the electron and hole are given separately in the insets for each applied external field. In the absence of the external fields (in the presence of any external field taken into consideration), the binding energy is a decreasing function of all the well sizes (in the weak confinement regime). This result is unavoidable. As the well sizes increase, although the electron and the hole are very well localized in the well, the probability of finding them in the same plane decreases since the spatial spreads of the electron and hole wavefunctions become very large (at the same time, the λ exciton extension also becomes larger), and therefore EBEs decrease due to the weakening of the Coulomb interaction between them. When the electric field is applied, due to the increment in the well width, the EBEs increase until reaching a maximum value, where the system has a quasi-two-dimensional character, and then they decrease again. It should be noted that this change in binding energy will occur at smaller *k* values than we considered here in the absence of any external field. This behavior depends on the confinement degrees of the electron and hole in the quantum well. For the small values of *k* parameter, due to penetration from the barriers of the electron and hole which have higher energy, the weakening of the Coulombic interaction between the electron and hole pair results in a decrease in the EBE. For the large values of *k* parameter, although the low-energy electron and hole are localized better in the well due to the weak confinement, the EBE decreases due to the probability of finding the electron and the hole in the same plane and the weakening of the Coulombic interaction between them. At sufficiently small k− values, EBE is almost independent of the electric field since geometric confinement is dominant. For the wells with large size, the binding energy becomes sensitive to the electric field. As seen in the inset in Figure 1a, the electric field that is applied in the *z*-direction causes the quantum wells in which the electron and hole are confined to bend in opposite directions. Thus, the electron on the left side and the hole on the right side of the wells are localized. EBE decreases since the Coulombic interaction between the electron and hole decreases with the electric field. The magnetic field provides extra confinement for the electron and hole; since the electron is lighter and more energetic than the hole, the magnetic field effect is more effective on the electron. As seen from the inset in Figure 1b, the magnetic field creates sharply parabolic confinement for the electron while the effect of its on the hole has not started yet, but the effect of the magnetic field on the hole will also become evident in the weaker confinement regime. Thus, the magnetic field leads to an increment in the EBE. The effect of the ILF on the EBE is as seen in Figure 1c. EBE for $\alpha_{0}=0$ is greater than those of $\alpha_{0}=5.0$ nm and $\alpha_{0}=10.0$ nm values. The bottom of the quantum wells and also energies for both the electron–hole pair shift towards higher energy values with the effect of ILF. As the ILF increases, the particles become localized in the wider wells, and the binding energy decreases due to the increase in λ, as seen in Table 1, and the weakening of the Coulomb interaction between the electron–hole pair. The ILF effect is very dominant at small well widths, not at large *L* values, contrary to the effect of the electric field. We want to emphasize that the values we used for the magnitudes of the electric, magnetic, and laser fields are perfectly plausible in low-dimensional systems such as QWs with dimensions in the range of those we considered in this study; see, for example, refs. [42,43], and references therein.

Figure 2a–c have the same arrangements as Figure 1a–c but the results correspond to the confinement potentials $\left(V_{2}\left(z_{e(h)}\right)>0\right)$ defined in Equation (Equation 3) (see the next section) for $\nu_{2}=-2$. The insets are not given because the variation of potentials in the presence of external fields are as in Figure 1a–c. For this potential, for the same *k* values being taken into account, the EBEs are greater than the results of $V_{1}\left(z_{e(h)}\right)$ potential in all applied external fields since the Coulomb interaction between the electron and hole is greater. The behavior of the binding energy observed in Figure 1a is not observed at the initial values of the *k* parameter in Figure 2a since the electron and hole are localized in QWs which have moderate well widths—not a very strong confinement regime. When the λ values for each well in Table 1 and Table 2 are checked, it can be seen that the exciton extension for the $V_{2}\left(z_{e(h)}\right)$ potential is smaller than the $V_{1}\left(z_{e(h)}\right)$ potential; therefore, the EBE is greater at the QW which has $V_{2}\left(z_{e(h)}\right)$ potential.

Figure 3a–c have the same arrangements as Figure 1a–c, but results correspond to the hyperbolic double QW which has the confinement potentials defined for $\nu_{3}=4$ in Equation (Equation 4) (see the next section). The dependence of the binding energy on the well dimensions is as in Figure 1a and Figure 2a, but because the effective length in the hyperbolic double QW is greater than those of the other single hyperbolic QWs, the spatial spread of electron and hole wavefunctions is quite high, and therefore EBE is lower. The variation in the EBE in the presence and absence of the electric field is larger than that of single QWs, and this variation increases as *k* increases. While the binding energy in the single QWs is sensitive to different electric field intensities at large *k* values, the EBE in double QWs is independent of different field intensities. As is known, applying an external electric field causes distortion of the symmetries of the heterostructures (see inset of Figure 3a). Depending on the distortion, the electron and hole gradually start to localize in the opposite directions of the well. As the electric field strength and also the well widths increase, the electron (hole) becomes completely localized in the left (right) well, which causes a large increase in the λ, as seen in Table 3. In this case, since the spatial spreads of the electron and hole wavefunctions increase more and more, the Coulombic interaction also decreases gradually, and a wavy decrease in EBE is observed. The reason for the wavy decrement is because the electron and the hole are localized gradually, not simultaneously, in different wells with the effect of the electric field depending on their effective masses. For double QW, the magnetic field effect on the EBE is given in Figure 3b. With the effect of the magnetic field, as seen in the inset, the double QW in which the electron is confined turns into a narrower double QW that bottom-shifts towards high energies, whereas the hole is still localized in a larger double QW. The hole will have this type of confinement potential at greater magnetic field intensities. While the magnetic field compresses the electron wavefunction, the spatial spread of the hole wavefunction is still large; thus, the Coulombic interaction between the particles is weak and the EBE decreases. In fact, as the magnetic field increases, an increase in EBE is expected, but the energies of the electron and hole shift towards higher energies with the magnetic field, so the EBE decreases due to the weakening of Coulomb interaction between the electron and the hole (increasing magnetic field creates a decrease in EBE since the negative Coulomb term in Equation (Equation 6) (see the next section) becomes too large). The ILF effect on the EBE for the hyperbolic-type double QW is given in Figure 3c. As seen in the inset, the effects of ILF on geometric confinement are interesting. With the ILF effect, while the central barrier turns into a well (wells turn into a barrier), the bottoms of the wells also shift upwards and EBE decreases as the Coulombic interaction between the electron–hole pair reduces. The wavy reduction is observed in EBE towards increasing k−values due to the degree of competition between the ILF and confinement effects.

From Equations (7) and (8) (see the next section), it can be inferred that we have chosen the exciton trial wavefunction as the product between the *z*-dependent one-particle ground-state wavefunctions of the confined electron and hole multiplied by an exponential function which depends on the relative radial coordinate and the λ variational parameter. The resulting product is multiplied by the inverse of λ. It is clear that when the *k* parameter is large enough (k→∞), and in the absence of the applied electric field, our results must converge exactly to the binding energy of the freed exciton in the GaAs bulk material. That is, to $E_{b}=1\;R_{eff}$ ($1\;R_{eff}=5.2$ meV). That is exactly the trend shown in our results in Figure 1a–c for F=0, B=0, and $\alpha_{0}=0$, respectively. The presence of the 1/λ term in the trial wavefunction helps us to solve the problem of having chosen a hydrogenic function dependent only on the relative radial coordinate, and for that reason, our results reach the correct limit when k=200 nm $=17.9\;a_{B}$ ($1\;a_{B}=11.2$ nm is the GaAs exciton effective Bohr radius). Many authors decide to use a three-dimensional hydrogenic trial function which gives an excellent account of the limits of weak confinement, as k→∞, and strong confinement, as k→0. When the electric field turns on, a polarization of the electron–hole pair appears. This polarization depends on the magnitude of the applied electric field. When *k* is large enough, the overlap between the electron and hole wavefunctions tends to zero, and leads to essentially zero binding energy, as shown, for example, by the trend in Figure 1a, Figure 2a, and Figure 3a for F=25 kV/cm when k=20 nm. Thus, we can conclude that our results are in complete agreement with what is expected to happen at the limit of the large values of the *k* parameter.

From Equation (Equation 8) (see the next section), it can be seen that the number λ is directly connected with the extension of the exciton wavefunction in the plane of the heterostructure. Considering that $1\;a_{B}=11.2$ nm, the extension of the wavefunction for $V_{1}$ potential, according to Table 1, is in the range $1.19\;a_{B}-1.38\;a_{B}$ for $L_{eff}=10$ nm, and in the range $1.26\;a_{B}-1.60\;a_{B}$ for $L_{eff}=15$ nm. In the case of the $V_{3}$ potential, due to the double structure considered, these λ values of the wavefunction extension are significantly higher and are in line with the highest $L_{eff}$ values that are considered. Large values of λ parameter mean more extended wavefunctions and consequently lower value of the electron–hole Coulomb interaction, which leads to a lower binding energy. The opposite case occurs when the λ parameter takes smaller values: the Coulomb interaction is strengthened and, consequently, the binding energy is higher.

Figure 4, Figure 5a,b and Figure 6 show the exciton absorption coefficients (EACs) for interband transitions as a function of the resonant photon energy for $V_{1}\left(z_{e(h)}\right)$, $V_{2}\left(z_{e(h)}\right)$, and $V_{3}\left(z_{e(h)}\right)$ confinement potentials, respectively. Results are in the absence and presence of the external electric, magnetic, and intense laser fields. The values of the effective well widths $\left(L_{eff}\right)$ for each potential are given in the figures to which they belong.

In the absence and the presence of all external fields, the EACs corresponding to interband transitions in the QW, which has potential $V_{1}\left(z_{e(h)}\right)$ for two different effective well widths, are given in Figure 4. The peak positions of EAC shift to the blue with increasing magnitudes for the electric and magnetic fields, while they shift to the red with increasing magnitudes for the ILF, for both $L_{eff}$ values. Furthermore, as $L_{eff}$ increases, the peak positions of EAC shift to red with decreasing magnitudes. As $L_{eff}$ increases, the ITE (ΔE) in the denominator of the last factor in Equation (Equation 13) (see the next section) decreases due to the decrement in energies of the electron and hole, causing these peaks to shift to lower photon energies. The decrease in amplitudes of EACs is the result of the increase in the exciton extension. In fact, as $L_{eff}$ (or *k* parameter) increases, the λ variational parameter increases. The increase in the λ parameter means an increase in the exciton extension; amplitudes of EACs decrease with increasing well widths and also applied external fields since EAC in Equation (Equation 13) (see the next section) is proportional to $1/\lambda^{2}$. To prove the accuracy of the explanations mentioned above, the effects of ITE and exciton extension on the peak positions and amplitudes of the EACs are numerically presented in Table 1, depending on the well sizes and the externally applied fields.

In Figure 5a,b, as the well width increases, the peak positions of the EACs shift towards the red. For both $L_{eff}$ values, the magnetic field and the ILF cause the blue-shift while the electric field causes the red-shift in the peak positions of EACs. The variations of the peak positions and amplitudes of EACs versus the well widths and also applied external fields are given in Table 2 numerically.

In the absence and the presence of the external fields, EACs for double hyperbolic QW, which has $V_{3}\left(z_{e(h)}\right)$ potential as a function of the resonant photon energy, are given in Figure 6 for two different effective lengths. Inset shows the variation of the potential under ILF. It should be noted that the values of $L_{eff}=10$ nm and $L_{eff}=15$ nm correspond to values of k=10 nm and k=20 nm for both $V_{1}\left(z_{e(h)}\right)$ and $V_{2}\left(z_{e(h)}\right)$ potentials, respectively. The values of $L_{eff}=30$ nm and $L_{eff}=50$ nm correspond to values of k=10 nm and k=20 nm for $V_{3}\left(z_{e(h)}\right)$ potential, respectively. There is no reason other than those explanations above for the variation of the peak positions and amplitudes of the EACs versus the resonant photon energy concerning the well sizes and the external fields. In addition to the explanations above, in the presence of ILF and in the wider double QW, the magnitude of the EAC increases, unlike the magnitudes of the other peaks and/or the previous cases. As seen in the inset of this figure, due to the ILF-induced change on the QW, both the electron and hole are localized in the well at the center, which causes the overlap integral between the electron and hole pair, and hence the magnitude of this peak, to increase. For example, the overlap integral is OI=2.011 and OI=0.919 for $L_{eff}=30$ nm. Since the overlap integral is usually OI≤1, the effect of λ parameter on the peak magnitudes was very dominant, but here, OI>1 and EAC in Equation (Equation 13) (see the next section) is proportional to ${\left(OI\right)}^{2}=|\langle \psi_{i}\left(z_{h}\right)|\psi_{f}\left(z_{e}\right)\rangle {|}^{2}$; hence, the magnitude of this peak increases.

Finally, we would like to add that very careful studies of interdiffusion effects in QWs were reported by Li and Weiss [44]. The interdiffusion process of the trivalent elements across the well–barrier interface was described via Fick’s second law. In that sense, we want to emphasize that, for example, for the potential given by Equation (Equation 2) (see the next section), when k=20 nm the obtained energy for the ground state electron is −208 meV, and that such energy value is obtained for a QW with abrupt barriers whose width is 15.8 nm. More detailed studies of the interdiffusion process for the structures given by Equations (2)–(4) (see the next section) are in progress and will be published elsewhere.

## 3. Materials and Methods

In the effective mass approximation, the exciton Hamiltonian for a confined electron–hole pair in a quantum well region under the effects of applied electric and magnetic fields is as follows:(1)$$H=\frac{1}{2\;m_{e}^{*}}{\left[{\vec{p}}_{e}+\frac{e}{c}\vec{A}\left({\vec{r}}_{e}\right)\right]}^{2}+V\left(z_{e}\right)+e\;F\;z_{e}+\frac{1}{2\;m_{h}^{*}}{\left[{\vec{p}}_{h}-\frac{e}{c}\vec{A}\left({\vec{r}}_{h}\right)\right]}^{2}+V\left(z_{h}\right)-e\;F\;z_{h}-\frac{e^{2}}{\varepsilon \;|{\vec{r}}_{e}-{\vec{r}}_{h}|}\;,$$
where $m_{e}^{*}\left(m_{h}^{*}\right)$ is the electron (hole) effective mass, ${\vec{p}}_{e}\left({\vec{p}}_{h}\right)$ is the electron (hole) momentum operator, and $\vec{A}(\vec{r_{e})}$ ($\vec{A}(\vec{r_{h})}$) is the electron (hole) vector potential associated with the magnetic field; note that here the magnetic field is applied perpendicular to the structure growth direction (*z*-direction), i.e., the magnetic field and vector potential have the forms of $\vec{B}=\left(B,0,0\right)$ and $\vec{A}=\left(0,B\;z,0\right)$. In addition, *e* is the absolute value of the electron charge, *c* is the speed of light in vacuum, ε is the QW dielectric constant, *F* is the strength of electric field, which is applied parallel to the structure growth direction, and $V(z_{e(h)})$ is the confinement potential for electron and hole in the *z*-direction. Before the ILF is applied, the functional forms of the confinement potentials which are formed using three different hyperbolic-type functions are as follows:(2)$$V_{1}\left(z_{e(h)}\right)=-V_{0}^{e(h)}\;\nu_{1}\;{\mathrm{sech}}^{2}\left(z/k\right)\;,$$
(3)$$V_{2}\left(z_{e(h)}\right)=-V_{0}^{e(h)}\;\nu_{2}\;\tanh^{2}\left(z/k\right)\;,$$
(4)$$V_{3}\left(z_{e(h)}\right)=-V_{0}^{e(h)}\;\nu_{3}\;\sinh^{2}\left(z/k\right)\;{\mathrm{sech}}^{4}\left(z/k\right)\;,$$
where $V_{0}^{e(h)}$ is the band discontinuity for electron (hole) ($V_{0}^{e}=228$ meV and $V_{0}^{h}=176$ meV. Note that these values correspond to an aluminum concentration value of x=0.3 in GaAs-Al${}_{x}$Ga${}_{1-x}$ As QWs), $\nu_{1}$, $\nu_{2}$, and $\nu_{3}$ are the parameters related to the well depth, and *k* is the parameter that determines the well width.

In the presence of the ILF, the laser-dressed potential is defined as below:(5)$$\langle V_{i}\left(z_{e(h)},\alpha_{0}\right)\rangle =\frac{\bar{\omega }}{2\;\pi }\int_{0}^{2\;\pi /\bar{\omega }}V_{i}^{e(h)}\left(z_{e(h)}+\alpha_{0}\;\sin\bar{\omega }t\right)\;\;dt\;,\;\left(i=1,2,3\right)\;,$$
where $\alpha_{0}\left(=e\;A_{0}/\mu \;\bar{\omega }\right)$ is the laser-dressing parameter for the exciton, with $A_{0}$ and $\bar{\omega }$ the magnitude of the vector potential and the angular frequency of the non-resonant ILF, respectively. For the dressed potentials defined in Equation (Equation 5), the details and the non-perturbative approach based on the Kramers–Henneberger translation transformation developed to describe the atomic behavior in intense high-frequency ILF can be found in Refs. [45,46]. For a given laser source, whose frequency is ν and output power is *I*, the following practical formulae are useful: $F_{0}$ (in kV/cm) $\approx 0.87\;\sqrt{I}/\sqrt[{4}]{\varepsilon }$ and $\alpha_{0}$ (in units of $a_{B}$, the effective Bohr radius)$\;\approx 7.31\;\varepsilon^{-5/4}\;\sqrt{I}/\nu^{2}$, for *I* in kW/cm${}^{2}$ and ν in THz [47].

By defining the electron–hole center of mass $\left(\vec{R}=\left(m_{e}^{*}\;{\vec{r}}_{e}+m_{h}^{*}\;{\vec{r}}_{h}\right)/\left(m_{e}^{*}+m_{h}^{*}\right)\right)$ and the relative coordinates $\left(\vec{r}={\vec{r}}_{e}-{\vec{r}}_{h}\right)$, and assuming that the motion of the center of mass is constant, the exciton Hamiltonian in cylindrical coordinates is obtained as follows:$$H=-\frac{\hbar^{2}}{2\;\mu }\left(\frac{\partial^{2}}{\partial \rho^{2}}+\frac{1}{\rho }\frac{\partial }{\partial \rho }+\frac{1}{\rho^{2}}\frac{\partial^{2}}{\partial \phi^{2}}\right)-\frac{\hbar^{2}}{2\;m_{e}^{*}}\frac{\partial^{2}}{\partial z_{e}^{2}}+V\left(z_{e},\alpha_{0}\right)+e\;F\;z_{e}+\frac{e^{2}\;B^{2}\;\hbar^{2}}{4\;\mu \;c^{2}\;R_{eff}^{2}}\;z_{e}^{2}$$
(6)$$-\frac{\hbar^{2}}{2\;m_{h}^{*}}\frac{\partial^{2}}{\partial z_{h}^{2}}+V\left(z_{h},\alpha_{0}\right)-e\;F\;z_{h}+\frac{e^{2}\;B^{2}\;\hbar^{2}}{4\;\mu \;c^{2}\;R_{eff}^{2}}\;z_{h}^{2}-\frac{e^{2}}{\varepsilon \;\sqrt{\rho^{2}+{\left(z_{e}-z_{h}\right)}^{2}}}\;,$$
where $\rho ={\left[{\left(x_{e}-x_{h}\right)}^{2}+{\left(y_{e}-y_{h}\right)}^{2}\right]}^{1/2}$ is the relative distance between the electron and hole in the (x−y) plane. Additionally, $\mu =m_{e}^{*}\;m_{h}^{*}/\left(m_{e}^{*}+m_{h}^{*}\right)$ is the reduced electron–hole mass and $R_{eff}=\mu \;e^{4}/2\;\varepsilon^{2}\hbar^{2}$ is the exciton effective Rydberg. Note that in this work we did not include the intense laser field effects on the diamagnetic terms in Equation (Equation 6), i.e., on the quadratic terms for $z_{e}$ and $z_{h}$.

It is important to note that the term associated with the vector potential that connects with the magnetic field for both particles (electron and hole) gives rise to quadratic terms in the coordinate of each particle, as shown by the fifth and ninth terms on the right side of Equation (Equation 6). This corresponds to parabolas that open to infinity as the *z*-coordinate of each particle increases. By applying Equation (Equation 5) to these terms, the dressed potential that is obtained corresponds exactly to the same parabola displaced vertically in the energy axis. In that case, the wavefunctions of each particle, the matrix elements in Equation (Equation 11), the energy differences between confined states, and the binding energy do not change. Thus, considering the dressed potential for a parabola does not change the results presented here at all. As for the electric-field-dependent terms, this corresponds to linear equations, as shown by the fourth and eighth terms on the right-hand side of Equation (Equation 6). In this case, applying Equation (Equation 5), the resulting dressed potential gives rise to the same original equation. It is for all this that Equation (Equation 5) only applies to the confinement potentials given by the second and fifth terms on the right-hand side of Equation (Equation 1).

For the excitonic structure, by choosing the wavefunction in the form of
(7)$$\Phi \left(\rho ,\phi ,z_{e},z_{h},\lambda \right)=\psi \left(z_{e}\right)\;\psi \left(z_{h}\right)\;\varphi \left(\rho ,\lambda \right)\;,$$
the Schrödinger equation in the form of $H\left(\rho ,\phi ,z_{e},z_{h}\right)\;\Phi \left(\rho ,\phi ,z_{e},z_{h},\lambda \right)=E\;\Phi \left(\rho ,\phi ,z_{e},z_{h},\lambda \right)$ for the Hamiltonian in Equation (Equation 6) is solved by using the diagonalization method combined with the variational method. Details of the method used are given in Ref. [48]. Here *E*, $\psi (z_{e})$, and $\psi (z_{h})$ are the total energy and the electron and hole wavefunctions in the *z*-direction, respectively. The wavefunction with λ variational parameter describing the exciton motion in the (x−y) plane is given by [40,49,50]
(8)$$\varphi \left(\rho ,\lambda \right)=\frac{1}{\lambda }\;{\left(\frac{2}{\pi }\right)}^{\frac{1}{2}}e^{-\;\frac{\rho }{\lambda }}\;\;.$$

The λ variational parameter also means the exciton spatial extension; $E_{e}$ and $E_{h}$ are the first subband energies of the electron and heavy-hole. After the subband energies and the expectation value of the Hamiltonian in Equation (Equation 6) are obtained, EBE is defined as follows:(9)$$E_{b}=E_{e}+E_{h}-\langle \Phi |H|\Phi \rangle {|}_{min,\;\lambda }\;\;.$$

It should be noted that the value of λ, which minimizes the energy, is determined numerically. In the dipole approximation, the linear absorption coefficient corresponding to the excitonic transition for the heavy-hole electronis given by [51]
(10)$$\alpha \left(\omega \right)=\frac{1}{V}\frac{4\;\pi^{2}\;e^{2}}{n\;c\;m_{0}^{2}\;\omega }\;\;\sum_{i,f}\;\;|\langle i|\hat{\varepsilon }\;\cdot \hat{p}{\left|f\rangle \right|}^{2}\;\;\delta \left(E_{f}-E_{i}-\hbar \;\omega \right)\;,$$
where ω is the incident photon angular frequency, $V=L_{x}\;L_{y}\;L_{z}$ is the volume of the sample, *n* is the material refractive index, $\hat{\varepsilon }$ is the incident radiation polarization vector, $\hat{p}$ is the electron momentum operator, and 〈i| and |f〉 are initial and final states of the system, respectively. In the previous equation, the following condition can be demonstrated: (11)$$\left|\langle i\right|\hat{\varepsilon }\;\cdot \hat{p}{\left|f\rangle \right|}^{2}=|\langle U_{V}|\hat{\varepsilon }\;\cdot \;\hat{p}|U_{C}{\rangle |}^{2}\;\;|\langle \psi_{i}\left(z_{h}\right)|\psi_{f}\left(z_{e}\right)\rangle {|}^{2}\;\delta_{K_{⊥,0}}\;L_{x}\;L_{y}\;{\left|g_{t}\left(\rho =0,\lambda \right)\right|}^{2},$$
where $L_{x}$ and $L_{y}$ are the lengths in free directions, and $K_{⊥,0}=k_{⊥C}-k_{⊥V}$ is the difference between the conduction and valence band electron and heavy-hole wave vectors, respectively. The Kronecker $\delta_{K⊥,0}$ symbol in Equation (Equation 11) dictates that, for the interband transitions, the electron momentum in the *x* and *y* directions are conserved. Additionally, the $g_{t}\left(\rho =0,\lambda \right)$ term indicates that only excitons that have amplitude different from zero can absorb light at ρ=0. For *z*-polarized incident radiation, and the allowed interband transitions between the first heavy-hole and the first electron subbands, the first factor on the right-hand side in Equation (Equation 11) satisfies the condition [52,53]
(12)$$|\langle U_{V}|\hat{\varepsilon }\;\cdot \;\hat{p}|U_{C}{\rangle |}^{2}=|\langle U_{V}|p_{z}|U_{C}{\rangle |}^{2}=\frac{m_{0}}{4}\;E_{p}\;,$$
where $E_{p}$ is the Kane matrix element. Thus, inputting Equation (Equation 12) into Equation (Equation 10), and considering the Lorentzian representation of the δ-Dirac function, for excitons in a two-dimensional QW, the absorption coefficient, for heavy-hole electron transition [40,54], reduces to
(13)$$\alpha \left(\omega \right)=\frac{1}{L_{eff}\;\lambda^{2}}\frac{e^{2}\;E_{p}}{n\;c\;m_{0}\;\omega }|\langle \psi_{i}\left(z_{h}\right)|\psi_{f}\left(z_{e}\right)\rangle {|}^{2}\;\;\frac{\Gamma }{{\left(E_{e}+E_{h}+E_{g}^{GaAs}-E_{b}-\hbar \;\omega \right)}^{2}+{\left(\Gamma /2\right)}^{2}}\;.$$
where $L_{eff}$ is the well width (it should be noted that effective length has the form of $L_{eff}=L_{W}+L_{b}+L_{W}$ for double hyperbolic QW with $V_{3}(z_{e(h)}$ potential), Γ is the broadening parameter of the Lorentzian function, $E_{g}^{GaAs}$ is the band gap corresponding to GaAs material, ℏω is photon energy required for the transition to conduction band from valence band, and $OI=\langle \psi_{i}\left(z_{h}\right)|\psi_{f}\left(z_{e}\right)\rangle$ is overlap integral between the electron and hole wavefunctions [55]. In this context, the term $\Delta E=E_{e}+E_{h}+E_{g}^{GaAs}-E_{b}$ in the denominator of the last factor in Equation (Equation 13) is the interband transition energy (ITE), which we defined as ΔE.

## 4. Conclusions

In the present study, we investigated the exciton binding energy and absorption coefficient for the interband transition corresponding to the first electron and first heavy-hole states in the single and double QWs which have different hyperbolic-type potentials under the effects of electric, magnetic, and intense laser fields. The results obtained show that the geometric shape of the structure and the applied external fields are very effective on the electronic and optical properties. Our results show the following: (*i*) The exciton binding energy is a decreasing function of the effective well width, because the spatial spreads of the wavefunctions of the electron–hole pair, the exciton extension, and overlap integral between them increase more, and these cause the weakening of the Coulomb interaction between electron and hole. (*ii*) As the effective well width increases, the peak positions of exciton absorption coefficients shift towards the red with decreasing magnitudes. (*iii*) Since the interband transition energy (ΔE) decreases due to the decrement in energies of the electron and hole with increasing the well size, the resonant peaks shift to the lower photon energies. (*iv*) The decrease in amplitudes of exciton absorption coefficients is the result of the increase in exciton extension. (*v*) As a result, the tunability of the excitonic effects is expected to be of importance in developing stable and high-efficiency nanoscale excitonic optoelectronic devices. Moreover, before the growth of any low-dimensional quantum structure, it will be very practical and instructive to create and study the single, double, or triple quantum wells by selecting the appropriate hyperbolic potential functions and their different combinations by using the relevant material parameters. Finally, we would like to emphasize that the results we obtained in this study are quite consistent with the results of similar studies on QWs [33,54,55,56]. In particular, we want to emphasize that the maximum binding energy values that we found in this investigation for the strong confinement regime (the situation in which the electron and hole wavefunctions overflow into the barrier regions) are in excellent agreement with the report of Belov and Khramtsov for excitons in narrow quantum wells [57]. Considering the effect of the electric field applied along the growth direction, the reduction of the QW effective gap, for the considered heterostructures and shown in the insets of Figure 1a and Figure 3a, is in agreement with the observed red-shifts in experimentally obtained and theoretically modeled photoluminescence spectra in single and double QWs (see, for example, de Dios-Leyva et al. and reference therein [42]).

## Figures and Tables

**Figure 1 ijms-23-11429-f001:**
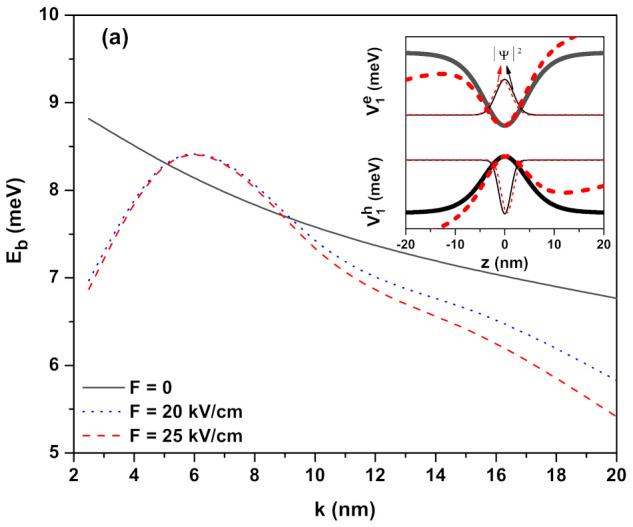
Heavy-hole exciton binding energy as a function of the *k* well width parameter for confinement potential $V_{1}\left(z_{e(h)}\right)=-V_{0}^{e(h)}\;\nu_{1}\;{\mathrm{sech}}^{2}\left(z/k\right)$ in the presence of the external fields: (**a**) electric field (B=0 and $\alpha_{0}=0$), (**b**) magnetic field (F=0 and $\alpha_{0}=0$), and (**c**) intense laser field (F=0 and B=0), respectively. The inset in each figure shows the changes in the confinement potentials (thick lines) and the square of the ground state wavefunction (thin lines) for the electron and hole and considering the lowest and highest value of the corresponding field in each panel. Results are for $\nu_{1}=1$.

**Figure 2 ijms-23-11429-f002:**
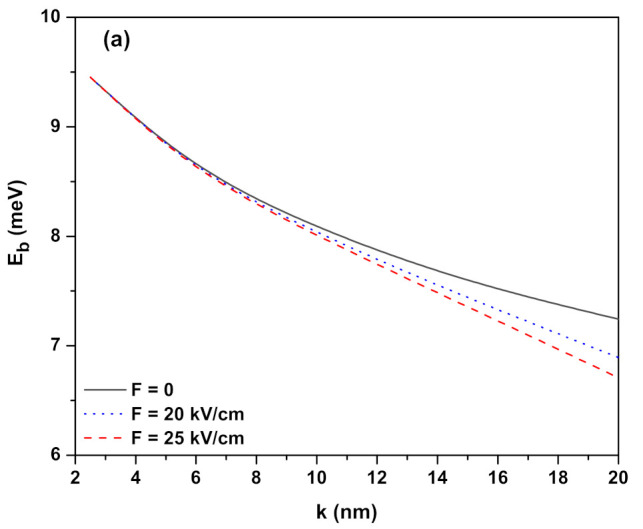
Heavy-hole exciton binding energy as a function of the well width parameter for confinement potential $V_{2}\left(z_{e(h)}\right)=-V_{0}^{e(h)}\;\nu_{2}\;\tanh^{2}\left(z/k\right)$ in the presence of the external fields: (**a**) electric field (B=0 and $\alpha_{0}=0$), (**b**) magnetic field (F=0 and $\alpha_{0}=0$), and (**c**) intense laser field (F=0 and B=0), respectively. The insets are not provided here because they are the same as in Figure 1a–c. Results are for $\nu_{2}=-2$.

**Figure 3 ijms-23-11429-f003:**
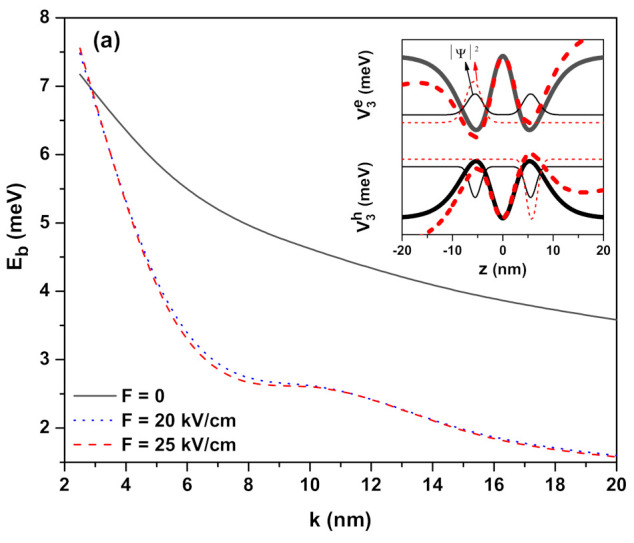
Heavy-hole exciton binding energy as a function of the well width parameter for confinement potential $V_{3}\left(z_{e(h)}\right)=-V_{0}^{e(h)}\;\nu_{3}\;\sinh^{2}\left(z/k\right)\;{\mathrm{sech}}^{4}\left(z/k\right)$ in the presence of the external fields: (**a**) electric field (B=0 and $\alpha_{0}=0$), (**b**) magnetic field (F=0 and $\alpha_{0}=0$), and (**c**) intense laser field (F=0 and B=0), respectively. The inset in each figure shows the changes in the confinement potentials (thick lines) and the square of the ground state wavefunction (thin lines) for the electron and hole, considering the lowest and highest values of the corresponding field in each panel. Results are for $\nu_{3}=4$.

**Figure 4 ijms-23-11429-f004:**
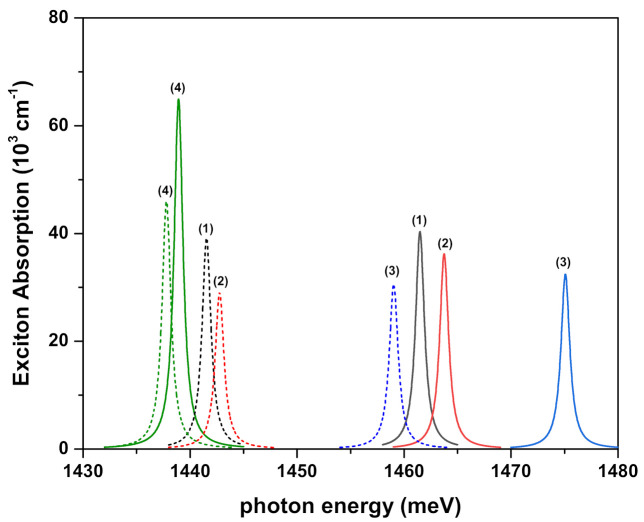
Exciton absorption coefficients for interband transitions as a function of the incident photon energy for confinement potential $V_{1}\left(z_{e(h)}\right)$ with $\nu_{1}=1$. Several combinations of the fields $\left(F,B,\alpha_{0}\right)$ (with *F* in kV/cm, *B* in Tesla, and $\alpha_{0}$ in nm) were considered and are indicated by labels 1, 2, 3, and 4: (0,0,0) (1), (25,0,0) (2), (0,10,0) (3), and (0,0,10) (4). Solid lines are for $L_{eff}=10$ nm, whereas dashed lines are for $L_{eff}=15$ nm.

**Figure 5 ijms-23-11429-f005:**
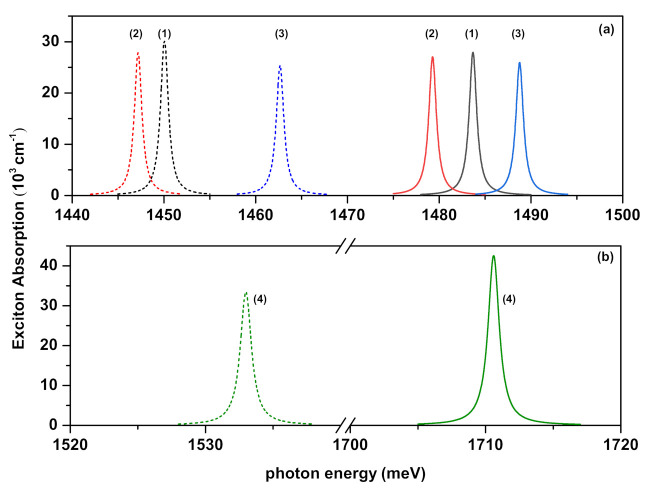
The exciton absorption coefficients for interband transitions as a function of the incident photon energy for confinement potential $V_{2}\left(z_{e(h)}\right)$, with $\nu_{2}=-2$. Several combinations of the fields $\left(F,B,\alpha_{0}\right)$ (with *F* in kV/cm, *B* in Tesla, and $\alpha_{0}$ in nm) were considered and are indicated by labels 1, 2, 3, and 4: in (**a**) the results are for (0,0,0) (1), (25,0,0) (2), and (0,10,0) (3). In (**b**) the results are for (0,0,10) (4). Solid lines are for $L_{eff}=10$ nm, whereas dashed lines are for $L_{eff}=15$ nm.

**Figure 6 ijms-23-11429-f006:**
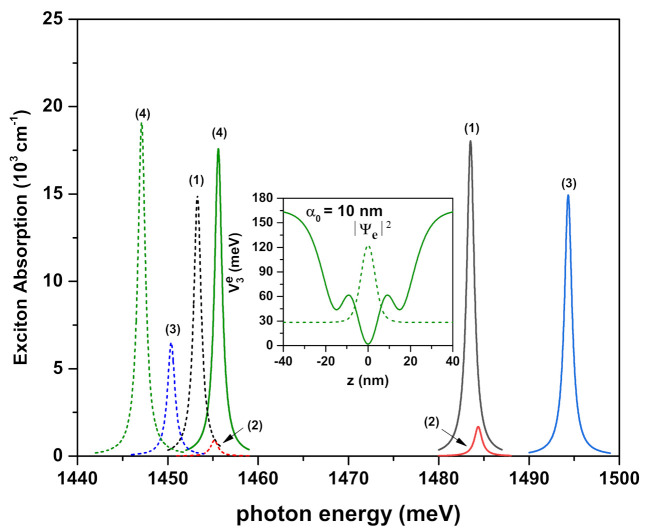
Exciton absorption coefficients for interband transitions as a function of the incident photon energy for confinement potential $V_{3}\left(z_{e(h)}\right)$ with $\nu_{3}=4$. Several combinations of the fields $\left(F,B,\alpha_{0}\right)$ (with *F* in kV/cm, *B* in Tesla, and $\alpha_{0}$ in nm) were considered and are indicated by labels 1, 2, 3, and 4: (0,0,0) (1), (25,0,0) (2), (0,10,0) (3), and (0,0,10) (4). Solid lines are for $L_{eff}=30$ nm, whereas dashed lines are for $L_{eff}=50$ nm. The inset shows the changes in the confinement potentials and the square of the electron ground state wavefunction for $\alpha_{0}=10$ nm.

**Table 1 ijms-23-11429-t001:** The variations of λ and ΔE parameters versus well sizes and external fields for $V_{1}\left(z_{e(h)}\right)$ potential.

$L_{\mathrm{eff}}$	*F*	*B*	$\alpha_{0}$	λ	$\Delta E=E_{e}+E_{h}+E_{g}^{\mathrm{GaAs}}-E_{b}$
(nm)	(kV/cm)	(Tesla)	(nm)	(nm)	(meV)
10	0	0	0	13.3321	1462
	25	0	0	14.1653	1464
	0	10	0	13.4987	1475
	0	0	10	15.4985	1439
15	0	0	0	15.1652	1442
	25	0	0	17.9983	1443
	0	10	0	14.1653	1459
	0	0	10	15.6652	1438

**Table 2 ijms-23-11429-t002:** The variations of λ and ΔE parameters versus well sizes and external fields for $V_{2}\left(z_{e(h)}\right)$ potential.

$L_{\mathrm{eff}}$	*F*	*B*	$\alpha_{0}$	λ	$\Delta E=E_{e}+E_{h}+E_{g}^{\mathrm{GaAs}}-E_{b}$
(nm)	(kV/cm)	(Tesla)	(nm)	(nm)	(meV)
10	0	0	0	13.1654	1481
	25	0	0	13.3321	1479
	0	10	0	12.9988	1489
	0	0	10	14.8319	1711
15	0	0	0	14.332	1450
	25	0	0	15.3318	1447
	0	10	0	13.832	1463
	0	0	10	14.8319	1533

**Table 3 ijms-23-11429-t003:** The variations of λ and ΔE parameters versus well sizes and external fields for $V_{3}\left(z_{e(h)}\right)$ potential.

$L_{\mathrm{eff}}$	*F*	*B*	$\alpha_{0}$	λ	$\Delta E=E_{e}+E_{h}+E_{g}^{\mathrm{GaAs}}-E_{b}$
(nm)	(kV/cm)	(Tesla)	(nm)	(nm)	(meV)
30	0	0	0	19.4987	1483.5
	25	0	0	32.8303	1484.4
	0	10	0	19.6648	1493
	0	0	10	14.4986	1456
50	0	0	0	22.4979	1453
	25	0	0	47.9955	1455
	0	10	0	29.4972	1450
	0	0	10	23.6645	1447

## Data Availability

No new data were created or analyzed in this study. Data sharing is not applicable to this article.

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
