# Peer review of "Theoretical Study of the Exciton Binding Energy and Exciton Absorption in Different Hyperbolic-Type Quantum Wells under Applied Electric, Magnetic, and Intense Laser Fields"

_ijms, 2022, doi:10.3390/ijms231911429_

Round 1
Reviewer 1 Report
The authors investigated the exciton binding energies and absorption coefficients for confined excitons in the single and double quantum wells which have different hyperbolic-type potentials under the effects of electric, magnetic, and intense laser fields.
The work is a complete and proper study that is quite timely in view of the numerous works on interlayer excitons in double quantum wells. The results are clean and clear and they are consistent with other works.
Therefore I suggest the publication of the paper.
Author Response
Journal: IJMS (ISSN 1422-0067)
Manuscript ID: ijms-1909181
Type: Article
Title: Theoretical Study of the Exciton Binding Energy and Exciton Absorption in Different Hyperbolic-Type QuantumWells under Applied Electric, Magnetic, and Intense Laser Fields
Authors: Melike Behiye Yücel, Huseyin Sari, Carlos Mario Duque, CARLOS ALBERTO DUQUE, Esin Kasapoglu
Section: Physical Chemistry and Chemical Physics
Collection: Feature Papers in Physical Chemistry and Chemical Physics
Referee 1
The Referee:
The authors investigated the exciton binding energies and absorption coefficients for confined excitons in the single and double quantum wells which have different hyperbolic-type potentials under the effects of electric, magnetic, and intense laser fields.
The work is a complete and proper study that is quite timely in view of the numerous works on interlayer excitons in double quantum wells. The results are clean and clear and they are consistent with other works.
Therefore I suggest the publication of the paper.
Our reply:
We want to thank the Referee for his/her very positive opinion about our work.
Reviewer 2 Report
Analytical models, including quasi-exactly-solvable models are highly interesting in many areas of physics, including the physics of semiconductors and semiconductor low dimensional structures.
This papers presents a family of quasy-exactly solvable model confinement potentials based on hyperbolic functions. Using these special kind of potentials, authors show the variational solution for semiconductor quantum well under effective mass approximation taking into account e-h interaction and external fields. As a result, 2D excitons localized in the proposed confinement potentials are analyzed.
While reading, I noticed a number of minor issues. Also, I have a couple of general comments.
Minor issues are:
1. line 103: "Exciton forms an excellent medium" - Excitons are not "medium", excitons are quasiparticles inside the medium, so this is slightly confusing
2. line 127, band discontinuity values - ? It depends on Al content
3. line 136 - 138. I defined as laser power, but measured in kV/cm^2. Most likely, the units should be kW/cm^2. But, in this case, this is not power (measured in Watts), but laser intensity.
4. I wonder why the intense laser field is included only in confinement potential and does not contribute to vector potential A and electric field F in Hamiltonian (1).
5. line 166. "… \Gamma which has Gaussian form… " - ? Eq. (13) contains usual Lorentzian line shape describing the homogeneous broadening of the exciton state. Why is it called Gaussian? Gaussian line shape is usually used to describe inhomogeneous broadening related, for example, to size dispersion in the ensemble of quantum dots.
6. line 168 "valance band" instead of "valence band", "I = <i|f>" - I is already used for laser power (or intensity?)
7. Insets in Figs 1, 3. Since there are no vertical axis breaks, Eg of AlGaAs barriers is interpreted to be 0.5 eV.
8. Inset in Figs 1(a), 3(a). Why electric field leads to linear increase in Eg with z? Electric field should just identically incline both Ec and Ev. Moreover, according to the plots, hole wavefunction tends to shift to the area with higher confinement.
General comments:
1. Exciton by its nature is multi-particle excitation. It is little bit confusing to see the discussion of excitons in terms of electrons and holes wavefunctions, especially in band diagrams (insets in Figs. 1, 3). In general, exciton just cannot be shown in usual band diagram since it is a single-particle picture. However, this is at the discretion of authors.
2. It would be nice to relate somehow the parameters of model potentials with some real physical properties of the quantum wells. Usually, such kind of hyperbolic potentials are used to describe the interdiffusion of the well and barrier materials (in this case, just the diffusion of Al to the well). For example, the well width parameter k can be related to the diffusion length (see, for example, J. Appl. Phys. 70, 1054 (1991)).
Also, I suppose that modern MBE grown QWs have very sharp interfaces and can be described with high accuracy with usual “square-well” approximation. So, this kind of smooth interfaces resulting from hyperbolic confinement can represent some kind of specially designed structures with controlled gradual change of Al content over the growth direction, or usual quantum well structures subjected to some high-temperature post-growth processing that activates the diffusion processes.
3. I have a strong doubts that the paper is suited to the aims and scope of the International Journal of Molecular Sciences and a topical collection "Feature Papers in Physical Chemistry and Chemical Physics". In my opinion, this is pure condensed matter physics and I recommend authors to publish the paper in some appropriate journal.
Author Response
Journal: IJMS (ISSN 1422-0067)
Manuscript ID: ijms-1909181
Type: Article
Title: Theoretical Study of the Exciton Binding Energy and Exciton Absorption in Different Hyperbolic-Type QuantumWells under Applied Electric, Magnetic, and Intense Laser Fields
Authors: Melike Behiye Yücel, Huseyin Sari, Carlos Mario Duque, CARLOS ALBERTO DUQUE, Esin Kasapoglu
Section: Physical Chemistry and Chemical Physics
Collection: Feature Papers in Physical Chemistry and Chemical Physics
Referee 2
The Referee:
Analytical models, including quasi-exactly-solvable models are highly interesting in many areas of physics, including the physics of semiconductors and semiconductor low dimensional structures.
This paper presents a family of quasi-exactly solvable model confinement potentials based on hyperbolic functions. Using these special kind of potentials, authors show the variational solution for semiconductor quantum well under effective mass approximation taking into account e-h interaction and external fields. As a result, 2D excitons localized in the proposed confinement potentials are analyzed.
While reading, I noticed a number of minor issues. Also, I have a couple of general comments.
Our reply:
We want to thank the Referee for his/her very positive opinion about our work. The observations and comments made by the Referee have stimulated us to improve the quality of our article.
Minor issues are:
The Referee:
- line 103: "Exciton forms an excellent medium" - Excitons are not "medium", excitons are quasiparticles inside the medium, so this is slightly confusing
Our reply:
We thank the Referee for his observation.
The text of the original version of the manuscript
“Exciton forms an excellent medium that can be used for signal processing, and simultaneously directly link to optical communication.”
has been replaced by the following text in the revised version of the manuscript:
“Exciton can be used for signal processing and simultaneously directly link to optical communication.”
The Referee:
- line 127, band discontinuity values - ? It depends on Al content
Our reply:
We thank the Referee for his observation.
The text of the original version of the manuscript
"(V0e =228 meV and $ V0h =176 meV)”
has been replaced by the following text in the revised version of the manuscript:
“(V0e =228 meV and $ V0h =176 meV. Note that these values correspond to an aluminum concentration value of x=0.3 in GaAs-AlxGa1-xAs QWs)”
The Referee:
- line 136 - 138.Idefined as laser power, but measured in kV/cm^2. Most likely, the units should be kW/cm^2. But, in this case, this is not power (measured in Watts), but laser intensity.
Our reply:
We would like to thank the Referee for his/her comment. The Referee is right, in the revised version of the manuscript, the typo has been amended.
The Referee:
- I wonder why the intense laser field is included only in confinement potential and does not contribute to vector potential A and electric field F in Hamiltonian (1).
Our reply:
We want to thank the Referee for his/her comment. After Eq. (6) of the revised version of the manuscript, we added the following comment:
“It is important to note that the term associated with the vector potential that connects with the magnetic field for both particles (electron and hole) gives rise to quadratic terms in the coordinate of each particle, as shown by the fifth and ninth terms on the right side of Eq. (6). This corresponds to parabolas that open to infinity as the $z$-coordinate of each particle increases. By applying Eq. (5) to these terms, the derred potential that is obtained corresponds exactly to the same parabola displaced vertically in the energy axis. In that case, the wavefunctions of each particle, the matrix elements in Eq. (11), the energy differences between confined states, and the binding energy do not change. So, considering the dressed potential for a parabola does not change the results presented here at all. As for the electric field dependent terms, this corresponds to linear equations, as shown by the fourth and eighth terms on the right hand side of Eq (6). In this case, applying Eq. (5), the resulting dressed potential gives rise to the same original equation. It is for all this that Eq. (5) only applies to the confinement potentials given by the second and fifth terms on the right hand side of Eq. (1).”
The Referee:
- line 166. "… \Gamma which has Gaussian form… " - ? Eq. (13) contains usual Lorentzian line shape describing the homogeneous broadening of the exciton state. Why is it called Gaussian? Gaussian line shape is usually used to describe inhomogeneous broadening related, for example, to size dispersion in the ensemble of quantum dots.
Our reply:
Before and after Eq. (13) we added the following two texts:
“So, using Eq. (12) into the Eq. (10), and considering the Lorentzian representation of the d-Dirac function, for excitons in a two-dimensional QW, the absorption coefficient, for heavy-hole-electron transition [44,50], reduces to”
“G is the broadening parameter of the Lorentzian function”
The Referee:
- line 168 "valance band" instead of "valence band", "I = <i|f>" - I is already used for laser power (or intensity?)
Our reply:
We want to thank the Referee for his/her comment. The manuscript has been amended.
The Referee:
- Insets in Figs 1, 3. Since there are no vertical axis breaks, Eg of AlGaAs barriers is interpreted to be 0.5 eV.
Our reply:
We thank the Referee for his/her observation. On the vertical axis of the insets of Figs. 1 and 3, we have removed the tick labels and ticks. We consider that they do not contribute any important information for the discussions. What we have wanted to show here is fundamentally the profiles of the electron and hole probability densities for their ground states.
The Referee:
- Inset in Figs 1(a), 3(a). Why electric field leads to linear increase in Eg with z? Electric field should just identically incline both Ec and Ev. Moreover, according to the plots, hole wavefunction tends to shift to the area with higher confinement.
Our reply:
At the end of the Conclusions section we have added the following comment with its corresponding reference:
“The reduction of the QW effective gap when considering the effect of a growth-direction applied electric field, for the considered heterostructures and shown in the insets of Figs. 1(a) and 3(a), is in agreement with the observed redshifts in experimentally obtained and theoretically modeled photoluminescence spectra in single and double QWs, see for example de Dios-Leyva \textit{et al.} and reference therein [49].”
We want to thank the Referee for having detected an-error that we had in the insets of Figs. 1(a) and 3(a). Now, in the revised versión of the manuscript, it is clear that the hole moves in the same direction as the electric field is applied, that is, in the direction in which the confining potential becomes smaller.
General comments:
The Referee:
- Exciton by its nature is multi-particle excitation. It is little bit confusing to see the discussion of excitons in terms of electrons and holes wavefunctions, especially in band diagrams (insets in Figs. 1, 3). In general, exciton just cannot be shown in usual band diagram since it is a single-particle picture. However, this is at the discretion of authors.
Our reply:
We ask the Referee to allow us to retain the model we have used to present how the electron and hole interact through the Coulomb interaction and which essentially depends on the location of the maxima of the probability densities of the two carriers. That ultimately gives information on the average distance between both carriers. That is the sense for using the insets in Figs. 1 and 3 and that help us in the discussions.
The Referee:
- It would be nice to relate somehow the parameters of model potentials with some real physical properties of the quantum wells. Usually, such kind of hyperbolic potentials are used to describe the interdiffusion of the well and barrier materials (in this case, just the diffusion of Al to the well). For example, the well width parameter k can be related to the diffusion length (see, for example, J. Appl. Phys. 70, 1054 (1991)).
Also, I suppose that modern MBE grown QWs have very sharp interfaces and can be described with high accuracy with usual “square-well” approximation. So, this kind of smooth interfaces resulting from hyperbolic confinement can represent some kind of specially designed structures with controlled gradual change of Al content over the growth direction, or usual quantum well structures subjected to some high-temperature post-growth processing that activates the diffusion processes.
Our reply:
At the end of the Results and Discussion section we have added the following paragraph:
“Finally, we would like to add that very careful studies of interdiffusion effects in QWs have been reported by Li and Weiss \cite{\Li1991}. The interdiffusion process of the trivalent elements across the well-barrier interface was described via the Fick's second law. In that sense, we want to emphasize that, for example, for the potential given by Eq. (2), when $k=20$\,nm the obtained energy for the ground state electron is $-208$\,meV and that such energy value is obtained for a QW with abrupt barriers whose width is $15.8$\,nm. More detailed studies of the interdiffusion process for the structures given by Eqs. (2-4) are in process and will be published elsewhere.”
The Referee:
- I have a strong doubt that the paper is suited to the aims and scope of the International Journal of Molecular Sciences and a topical collection "Feature Papers in Physical Chemistry and Chemical Physics". In my opinion, this is pure condensed matter physics and I recommend authors to publish the paper in some appropriate journal.
Our reply:
We hope that the Referee finds our responses to his/her observations and comments satisfactory and that he/she considers the modified version of our manuscript to be suitable for publication in the International Journal of Molecular Sciences.

Round 2
Reviewer 2 Report
Dear Authors!
I would like to thank you for careful addressing all my comments!
I think, the manuscript can be accepted for publication.